# Structure of Venezuelan equine encephalitis virus in complex with the LDLRAD3 receptor

Katherine Basore[1], Hongming Ma[2], Natasha M. Kafai[1,2], Samantha Mackin[1,2], Arthur S. Kim[1,2], Christopher A. Nelson[1], Michael S. Diamond[1,2,3,4 ✉] & Daved H. Fremont[1,3,4,5 ✉]

LDLRAD3 is a recently defined attachment and entry receptor for Venezuelan equine encephalitis virus (VEEV)[1], a New World alphavirus that causes severe neurological disease in humans. Here we present near-atomic-resolution cryo-electron microscopy reconstructions of VEEV virus-like particles alone and in a complex with the ectodomains of LDLRAD3. Domain 1 of LDLRAD3 is a low-density lipoprotein receptor type-A module that binds to VEEV by wedging into a cleft created by two adjacent E2–E1 heterodimers in one trimeric spike, and engages domains A and B of E2 and the fusion loop in E1. Atomic modelling of this interface is supported by mutagenesis and anti-VEEV antibody binding competition assays. Notably, VEEV engages LDLRAD3 in a manner that is similar to the way that arthritogenic alphaviruses bind to the structurally unrelated MXRA8 receptor, but with a much smaller interface. These studies further elucidate the structural basis of alphavirus–receptor interactions, which could inform the development of therapies to mitigate infection and disease against multiple members of this family.

Alphaviruses are enveloped, arthropod-transmitted single-stranded positive-sense RNA viruses that infect many vertebrate hosts, including humans, horses, rodents, birds and fish[2]. Alphaviruses can be categorized on the basis of their clinical syndromes: arthritogenic alphaviruses, such as chikungunya (CHIKV), Ross River, Sindbis (SINV) and O'nyong-nyong, cause arthritis, polyarthralgia and musculoskeletal-associated diseases; encephalitic alphaviruses, including Venezuelan (VEEV), Eastern (EEEV) and Western (WEEV) equine encephalitic viruses, cause meningitis, encephalitis and long-term neurological sequelae in survivors. The global distribution of alphaviruses has increased in recent decades owing to international travel, expansion of mosquito vectors, deforestation and urbanization[3].

Alphaviruses enter host cells through receptor-mediated endocytosis[4]. Within the low-pH endosomal compartment, the virion envelope rearranges to enable membrane fusion and nucleocapsid penetration into the cytoplasm[5]. The 12-kilobase alphavirus RNA genome is released after capsid disassembly and is translated from two open reading frames. The structural proteins (capsid, envelope glycoprotein (E)3, E2, 6K and E1) undergo processing and modification in the endoplasmic reticulum–Golgi network. The E2 and E1 proteins facilitate binding to entry factors and subsequent membrane fusion[6–9]. The E3 protein is essential for the proper folding of p62 (a precursor to E2) and the formation of the p62–E1 heterodimer[10,11] but is cleaved by furin-like proteases during maturation[12]. Mature E2–E1 heterodimers assemble into trimeric spikes at the plasma membrane before budding and release of the virion from the host cell[13]. The 70-nm-diameter mature alphavirus virion comprises 240 E2–E1 heterodimers that are arranged into 80 trimeric spikes with $T = 4$ icosahedral symmetry[14–16]. Twenty of these trimeric spikes sit on the icosahedral three-fold (i3) symmetry axes, and the other 60 spikes sit on the quasi-three-fold (q3) axes.

Low-density lipoprotein receptor class A domain-containing 3 (LDLRAD3) was recently identified as an attachment and entry receptor for VEEV and shown to be essential for optimal infection in cell culture and pathogenesis in mice[1]. LDLRAD3 is a conserved yet poorly characterized cell-surface protein that is expressed in neurons, epithelial cells, myeloid cells and muscle, the endogenous ligand(s) of which remain unknown. Biolayer interferometry experiments established that domain 1 (D1) of LDLRAD3 (LDLRAD3(D1)) binds directly to VEEV, and anti-LDLRAD3 antibodies and LDLRAD3(D1)–Fc fusion proteins block VEEV attachment and infection of cells. Only VEEV uses LDLRAD3 as a receptor, as EEEV, WEEV and other distantly related alphaviruses do not bind to it. How LDLRAD3 engages VEEV, and why only VEEV binds to LDLRAD3 remain unclear. We set out to address these questions using structural, genetic and biophysical approaches.

## Cryo-EM structure of LDLRAD3(D1) bound to VEEV

Mammalian-cell-expressed soluble LDLRAD3(D1) was produced in Expi293 cells[1]. Cryo-electron micrographs of VEEV virus-like particles (VLPs)[17] with or without bound LDLRAD3(D1) were acquired using a 300 kV Titan Krios system equipped with a Gatan K2 detector (Extended Data Fig. 1a and Supplementary Table 1). Single-particle analysis with imposed icosahedral symmetry yielded reconstructions at resolutions of 4.2 Å and 4.3 Å for the apo and complexed structures, respectively (Fig. 1a, b and Extended Data Fig. 1b). Two-hundred and forty molecules of LDLRAD3(D1) bound to sites on VEEV VLP (100% saturation), each one wedged into a cleft formed between two adjacent E2–E1 heterodimers

[1]Department of Pathology & Immunology, Washington University School of Medicine, St Louis, MO, USA. [2]Department of Medicine, Washington University School of Medicine, St Louis, MO, USA. [3]Department of Molecular Microbiology, Washington University School of Medicine, St Louis, MO, USA. [4]The Andrew M. and Jane M. Bursky Center for Human Immunology and Immunotherapy Programs, Washington University School of Medicine, St Louis, MO, USA. [5]Department of Biochemistry & Molecular Biophysics, Washington University School of Medicine, St Louis, MO, USA. ✉e-mail: diamond@wusm.wustl.edu; fremont@wustl.edu

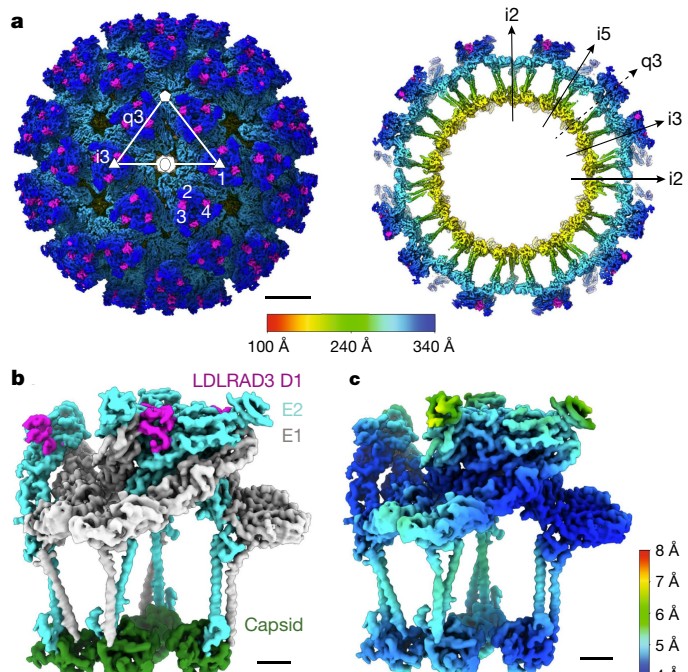

**Fig. 1 | Cryo-EM reconstruction of VEEV VLPs in complex with LDLRAD3(D1).** **a**, Coloured surface representation (left) and equatorial cross-section (right) of VEEV VLPs + LDLRAD3(D1). The surfaces are coloured by radial distance in Å, with the density of LDLRAD3 coloured magenta. The white triangle indicates one icosahedral asymmetric unit. The five-fold (i5), three-fold (i3) and two-fold (i2) icosahedral axes of symmetry are indicated by a pentagon, triangles and an oval, respectively. Trimeric spikes are labelled 'i3' if coincident with the i3 axes and 'q3' if on a quasi-three-fold axis. The black arrows indicate the directions of icosahedral symmetry axes (i2, i3, q3 and i5). Scale bar, 100 Å. **b**, **c**, Paired electron density of one asymmetric unit of the VEEV–LDLRAD3 complex, coloured by protein: E1 (grey), E2 (cyan), capsid (forest green) and LDLRAD3(D1) (magenta) (**b**) or by local resolution (**c**). Scale bars, 20 Å.

within each trimeric spike (Fig. 1c). This cleft widens slightly when D1 of LDLRAD3 is bound (Supplementary Video 1). Local resolution estimation performed in RELION revealed heterogenous resolution; the capsid proteins and membrane proximal regions of the E2–E1 heterodimers were best resolved (about 4 Å) and the membrane distal regions and LDLRAD3(D1) were less-well resolved (about 5–6 Å) (Fig. 1d). To avoid under- and over-sharpening of the reconstructions by conventional global *B*-factor correction, post-processing was performed using Deep-EMhancer[18]. This resulted in improved continuity and reduced noise in the density (Extended Data Fig. 1c). The visibly clear tracing of the carbon backbone simplified subsequent model building.

## Atomic model building and refinement

LDLRAD3(D1) was identified as an LDL receptor type A (LA) domain by the Pfam database[19]. LA domains are approximately 40 amino acids in length and contain 6 disulfide-bound cysteine residues and a cluster of conserved acidic residues that coordinate calcium ions (Fig. 2a). The LA domain architecture is well characterized with over 200 structures in the Protein Data Bank (PDB), revealing a highly conserved fold. The initial model of LDLRAD3(D1) was built from its primary amino acid sequence by threading using the SWISS-MODEL server[20] with multiple high-resolution crystal structures of related LA domains as templates. The starting coordinates of the VEEV VLP structural proteins came from a previously built model of the same VEEV strain (PDB: 3J0C; ref. [21]). Both models were docked into the DeepEMhancer modified electron density of the asymmetric unit and underwent manual and computational

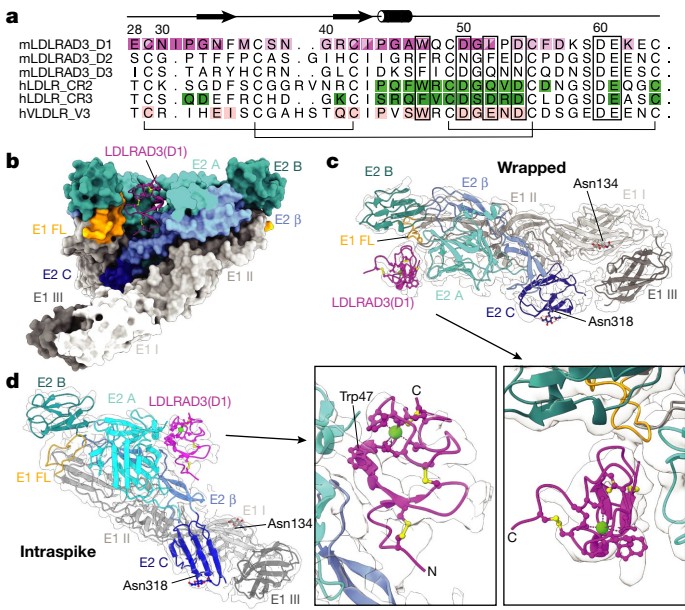

**Fig. 2 | Atomic model of LDLRAD3 interactions with VEEV. a**, Structure-based sequence alignment with the labelled secondary structure of various LA domains, including mouse (m) LDLRAD3 domains 1–3, human (h) LDLR CR2 and CR3 (PDB: 5OYL and 5OY9, respectively[31]), and human VLDLR-V3 (PDB: 3DPR; ref. [33]). Contact residues of LDLRAD3(D1) to the wrapped and intraspike VEEV E2–E1 heterodimers are shaded dark and/or light purple, respectively. Contact residues of the cysteine-rich domain 2 of LDLR (LDLR-CR2) and LDLR-CR3 to glycoprotein G of VSV (VSV G) are shaded green and contact residues of VLDL receptor module 3 (VLDLR-V3) to viral protein 1 (VP1) of human rhinovirus 2 (HRV2) are shaded pink, as determined by PDBePISA (www.ebi.ac.uk/pdbe/pisa/) (Fig. 4c–e). The brackets and rectangles indicate residues that form disulfide bonds and coordinate calcium, respectively. The figure was prepared using ALINE[33]. **b**, Ribbon diagram of LDLRAD3(D1) and surface representation of its wrapped and intraspike E2–E1 heterodimers. LDLRAD3(D1) and VEEV E2–E1 are coloured by domain. LDLRAD3(D1) (purple); chain E1: DI (light grey), DII (medium grey), DIII (dark grey) and fusion loop (FL) (orange); chain E2: A domain (cyan), β-linker (medium blue), B domain (dark cyan) and C domain (blue). The disulfide bonds and calcium ion in the ribbon diagram are coloured yellow and green, respectively. **c**, **d**, Paired isolated views of electron density and a model of LDLRAD3(D1) and its wrapped (**c**) or intraspike (**d**) heterodimers. Wrapped refers to the E2–E1 heterodimer, the fusion loop of which is covered by LDLRAD3. Intraspike refers to the heterodimer adjacent to the wrapped heterodimer but within the same trimeric spike. The naming convention is consistent with previous alphavirus–receptor structural studies[23]. The arrows indicate the regions that are magnified in the insets, which contain views of LDLRAD3(D1). Proteins are coloured by domain as described in **b**. *N*-linked glycans are shown as balls and sticks and coloured by heteroatom. The disulfide bonds and calcium ion are coloured yellow and green, respectively.

real-space refinement using COOT[22] and PHENIX[23] (Methods), with LDLRAD3(D1) unambiguously oriented with the N terminus proximal to the core of the virus particle (Fig. 2b–d and Supplementary Table 2).

The resultant model shows the domains and residues of the VEEV E2–E1 heterodimers at the LDLRAD3-binding interface. The two E2–E1 heterodimers at each binding site were termed 'wrapped' and 'intraspike', as described previously for the structure of CHIKV[23] in complex with its MXRA8 receptor. At the wrapped heterodimer interface, LDLRAD3 engages domains A and B of E2 (residues 24–28, 70–71, 166–199, 176–177 and 223) and the fusion loop in E1 (residues 85 and 87–92). On the intraspike heterodimer, LDLRAD3 interacts with domain A and the β-linker of E2 (residues 5, 63–64, 79, 92–95, 148, 153–159 and 262–267; Fig. 2c, d, Extended Data Figs. 2 and 3, and Supplementary Table 3). The binding interface is around 900 Å² with equal contributions from the interfaces of the wrapped and intraspike E2–E1 heterodimers.

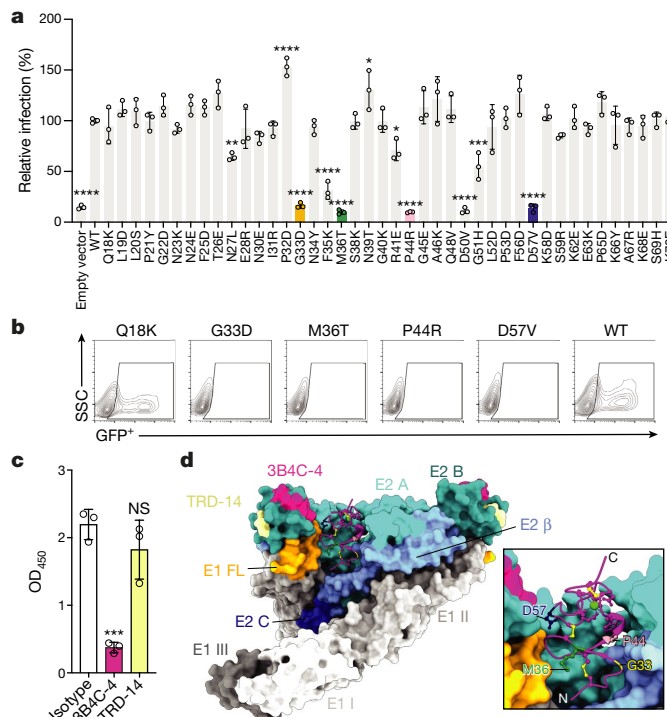

**Fig. 3 | Experimental assessment of the VEEV–LDLRAD3 model.**
**a**, **b**, Δ*B4galt7*Δ*Ldlrad3* Neuro2a cells complemented with WT *Ldlrad3* or the indicated mutants of *Ldlrad3* were inoculated with chimeric SINV–VEEV–GFP viruses (IAB strain TrD). Subsequently (7.5 h later), the infection levels were assessed (**a**) by monitoring GFP expression using FACS analysis (**b**). Data are mean ± s.d. of three experiments performed in technical duplicate. Each data symbol is the average of a technical duplicate from one experiment. $n = 3$. Statistical analysis was performed using one-way analysis of variance (ANOVA); *$P = 0.0317$ (N39T) or 0.0453 (R41E), **$P = 0.0054$, ***$P = 0.001$, ****$P < 0.0001$. The *Ldlrad3* transgene contains an N-terminal Flag tag downstream of the signal sequence for monitoring plasma membrane expression by flow cytometry (Extended Data Fig. 4). **c**, Competition binding analysis of LDLRAD3(D1)–human Fc and anti-VEEV mouse monoclonal antibodies (3B4C-4 and TRD-14) by ELISA. VEEV VLPs were incubated with anti-VEEV monoclonal antibodies (3B4C-4 and TRD-14) or anti-HCV H77.39 isotype control followed by detection with LDLRAD3(D1)–human Fc. Data are mean ± s.d. of three experiments performed in technical triplicate. Each data symbol is the average of a technical triplicate from one experiment. $n = 3$. Statistical analysis was performed using one-way ANOVA; ***$P = 0.0004$; NS, not significant. $OD_{450}$, optical density at 450 nm. **d**, Ribbon diagram of LDLRAD3(D1) and a surface representation of its wrapped and intraspike E2–E1 heterodimers with labelled epitopes of anti-VEEV mouse monoclonal antibodies (3B4C-4 and TRD-14) and labelled positions of LDLRAD3 mutants. Proteins are coloured by domain. LDLRAD3(D1) (purple); chain E1: DI (light grey), DII (medium grey), DIII (dark grey) and fusion loop (orange); chain E2: A domain (cyan), β-linker (medium blue), B domain (dark cyan) and C domain (blue). Inset: magnified view of the LDLRAD3(D1) ribbon diagram. The positions of mutations that resulted in reduced VEEV infection (G33 (light yellow), M36 (dark green), P44 (light pink) and D57 (dark blue)) are shown as balls and sticks. The N and C termini are labelled, and the disulfide bonds and calcium ion are coloured yellow and green, respectively.

The LDLRAD3 residues at the interaction interface that contribute to binding of the wrapped heterodimer are 29, 34, 36, 38–44, 47, 54–57 and 62. At the intraspike heterodimer interface, residues 28–34, 42–47 and 50–52 form contacts (Supplementary Table 3).

## Functional assessment of the atomic model

To assess our model, non-conservative point mutations were introduced throughout D1 of LDLRAD3 and used for complementation experiments in mouse Neuro2a cells lacking *Ldlrad3* (Δ*Ldlrad3*) and glycosaminoglycan (Δ*B4galt7*) expression[1]; we performed these experiments in cells lacking glycosaminoglycans to minimize background infection, as some alphaviruses also attach to cells through engagement of heparan sulfate moieties[17,24,25]. Wild-type (WT) LDLRAD3 and single point mutants of LDLRAD3 were transduced into Δ*B4galt7*Δ*Ldlrad3* Neuro2a cells, which were then inoculated with a chimeric, attenuated SINV–VEEV virus that expresses the structural genes of VEEV Trinidad Donkey (TrD) such that the screen could be performed using flow cytometry at a lower biosafety containment level (BSL2) yet with VEEV structural proteins from a pathogenic subtype IAB isolate. Whereas most mutant forms of LDLRAD3 promoted SINV–VEEV infection, several (including G33D, M36T, P44R and D57V) did not support infection even though the proteins were expressed on the cell surface at similar levels compared to the WT form of LDLRAD3 (Fig. 3a, b and Extended Data Fig. 4). The residues identified as loss-of-function for infectivity all sit in a pocket of LDLRAD3 that supports direct contact with residues of E2–E1 in both the wrapped and intraspike heterodimers (Supplementary Table 3). Several other mutations in LDLRAD3(D1) that correspond to contact residues (including P32D, N39T, A46K and F56D) appear to show slight increases in infectivity with normal surface expression patterns. Although further studies are required, these changes could enhance the affinity of VEEV binding.

Several years ago, a high-resolution cryo-electron microscopy (cryo-EM) structure of Fab fragments of the 3B4C-4 mouse monoclonal antibody bound to VEEV was published[26]. 3B4C-4 binds to the tip of the E2 B domain[27] and inhibits cellular attachment and entry of VEEV[28]. As the principal binding footprint (S177, V179, S180, L181, S184, T214, N216 and K223)[26] of this monoclonal antibody is proximal to the LDLRAD3-binding site, we tested whether 3B4C-4 could inhibit binding to LDLRAD3 using a competition binding enzyme-linked immunosorbent assay (ELISA). The 3B4C-4 monoclonal antibody was prebound to VEEV-VLP-coated plates before addition of the LDLRAD3(D1)–human Fc fusion protein. Notably, 3B4C-4 markedly inhibited LDLRAD3(D1) binding, whereas another anti-VEEV monoclonal antibody (TRD-14), which maps to a distinct epitope on the E2 B domain (G203, G204 and T205; N. Kafai and M. Diamond, unpublished data), did not compete for binding (Fig. 3c). A structural comparison of the monoclonal antibody epitopes on the E2 B domain revealed that 3B4C-4 binds to residues that are immediately adjacent to the LDLRAD3-binding site, probably resulting in steric hindrance (Fig. 3d). By contrast, the TRD-14 epitope is located at the distal end of the E2 B domain.

## D2 does not contribute to VEEV binding

D1 of LDLRAD3 is necessary and sufficient to support infection by VEEV[1], but it remains unclear whether D2 also contributes to VEEV binding. To evaluate this question, we expressed soluble LDLRAD3(D1+D2) in Expi293 cells (Extended Data Fig. 5a). Electron micrographs of VEEV VLPs with or without bound LDLRAD3(D1+D2) were acquired using a 300 kV Titan Krios system equipped with a Falcon 4 detector (Supplementary Table 1). Single-particle analysis with imposed icosahedral symmetry yielded a reconstruction at 5.0 Å (Extended Data Fig. 5b). The electron density of D2 of LDLRAD3 was weak and projected away from VEEV (Extended Data Fig. 5c). Binding of purified LDLRAD3(D1+D2) to captured VLPs by surface plasmon resonance yielded a monovalent affinity of approximately 50 nM that was similar to LDLRAD3(D1) (Extended Data Fig. 5d). On the basis of this structural and biophysical analysis, and previous functional data[1], D2 of LDLRAD3 does not appreciably contribute to VEEV binding or infection.

Cell culture infection experiments with mouse and human cells and in vivo pathogenesis studies in mice defined LDLRAD3 as a cell-surface receptor for VEEV that is required for optimal infectivity and induction of encephalitis in mice[1]. Here, our single-particle cryo-EM analyses of LDLRAD3 and VEEV VLPs provide structural insights into how VEEV

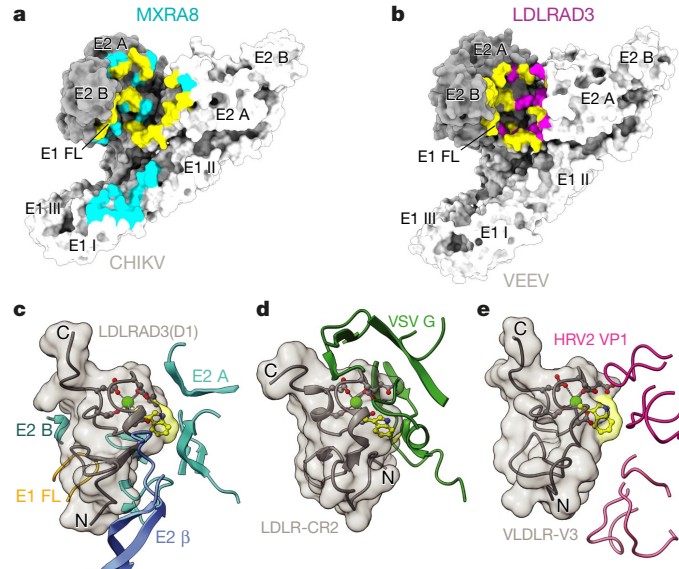

**Fig. 4 | Comparisons of VEEV–LDLRAD3 with other virus–receptor complexes. a**, Surface representation of the wrapped (dark grey) and intraspike E2–E1 (light grey) heterodimers of CHIKV, labelled by domain and coloured by determinants of MXRA8 receptor binding. Positions of determinants specific to MXRA8 are coloured cyan; positions shared with LDLRAD3 are yellow. The MXRA8 binding interface is about 2,100 Å². **b**, Surface representation of the wrapped (dark grey) and intraspike E2–E1 (light grey) heterodimers of VEEV, labelled by domain and coloured by determinants of LDLRAD3 receptor binding. Positions of determinants specific to LDLRAD3 are coloured magenta; positions shared with MXRA8 are coloured yellow. The LDLRAD3(D1) binding interface is about 900 Å². **c–e**, Paired, ribbon and surface diagrams of virus–LDLR structures. The calcium ions and tryptophan residues are coloured light green and yellow, respectively. **c**, Fragments of domains of the wrapped and intraspike E2–E1 heterodimers of VEEV (fusion loop of E1 (orange), domain A of E2 (cyan), β-linker of E2 (medium blue), domain B of E2 (dark cyan)) at the interface of LDLRAD3(D1) (grey). After VEEV binding, 29.0% of the solvent-accessible surface area (SASA) of LDLRAD3(D1) is lost. **d**, Fragments of VSV G (green) at the interface of LDLR-CR2 (grey) (PDB: 5OYL; ref. [31]). After VSV binding, 25.3% of the SASA of LDLR-CR2 is lost. **e**, Loops from two different copies of viral protein 1 (VP1; light and dark pink) from human rhinovirus 2 (HRV2) engage VLDLR-V3 (grey) (PDB: 3DPR; ref. [32]). After HRV2 binding, 13.1% of the SASA of VLDLR-V3 is lost.

engages with LDLRAD3 to facilitate interactions with target cells. We observed a network of quaternary protein–protein interactions with D1 of LDLRAD3 engaging two E2–E1 heterodimers within one trimeric spike. The specific binding determinants that we observed are supported by structure-guided mutations that we introduced into LDLRAD3, and binding competition studies with LDLRAD3(D1) and a neutralizing monoclonal antibody against VEEV that engages the top of the E2 B domain and directly blocks virus attachment. Our structures indicate that D1 of LDLRAD3 can bind with full occupancy at four distinct sites in the icosahedral asymmetric unit of the mature VEEV VLP.

VEEV binds to LDLRAD3 in a manner that is notably similar to the binding of CHIKV to its receptor MXRA8, which consists of two immunoglobulin-related folds[23,29,30] (Fig. 4a, b). Although LDLRAD3 and MXRA8 have similar sites of virion engagement, LDLRAD3 forms a significantly smaller interface (about 900 Å² versus about 2,100 Å²) even though the monovalent affinity of virus–receptor binding is similar[23] (Extended Data Fig. 5d). Inspection of the contact residues indicates that LDLRAD3 makes greater use of hydrophobic residues to bind to VEEV compared with the use of hydrophobic residues by MXRA8 when binding to CHIKV (about 40% versus about 24% of interface residues, respectively). Approximately 65% of the receptor contact positions

on VEEV spikes are shared with CHIKV (Extended Data Figs. 2 and 3). Both receptors effectively shield the hydrophobic fusion loop from solvent access, and all seven of the VEEV E1 contact residues are conserved with CHIKV E1. We speculate that the common positioning of these receptors near the fusion loop might function to modulate viral fusion during endocytosis. However, the primary contact residues used by LDLRAD3 and MXRA8 are not conserved; notably, MXRA8 has a substantial number of histidine residues (7% of the ectodomain) and LDLRAD3(D1) has no histidine residues.

The distinct receptor specificities of VEEV and CHIKV can probably be explained by the low level of sequence conservation of the E2-binding residues (26% of 35 LDLRAD3 contact positions). Our structural analysis also suggests why LDLRAD3 is a receptor for VEEV but not for WEEV and EEEV—other related encephalitic alphaviruses. Other than the aforementioned conserved contact site in the E1 fusion loop (100% conservation for 7 residues), the receptor determinants in E2 of VEEV generally are not conserved in WEEV and EEEV (17% and 23% identity, respectively, for 35 contact residues; Extended Data Figs. 2 and 3). By contrast, these determinants are essentially conserved among VEEV complex members, which probably explains why LDLRAD3 supports infection of all of the VEEV strains (IAB, IC and ID) that we tested[1].

LDL-receptor family members mediate the entry of several viruses belonging to different families. High-resolution structures have been solved for LDL receptor (LDLR) LA domains in complex with vesicular stomatitis virus (VSV) and human rhinovirus (HRV)[31,32]. Notably, an unrelated negative stranded rhabdovirus (VSV) and non-enveloped picornavirus (HRV) engage the same tryptophan residue near the calcium-binding site of the conserved cysteine-rich domain that also is a major contact for LDLRAD3 (Trp47) (Figs. 2a, Fig. 4c–e). Thus, evolutionarily distinct viruses have evolved similar structural strategies for engaging related members of a protein superfamily to enable entry into target cells. As such, it is plausible that structure-guided design of small-molecule inhibitors could prevent entry of viruses from multiple families.

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

# Methods

## Recombinant LDLRAD3 protein generation and purification

Monomeric LDLRAD3 ectodomain constructs were prepared as previously described[1]. In brief, mouse LDLRAD3(D1) (residues 18–70) and LDLRAD3(D1+D2) (residues 18–112) were cloned into the pCDNA3.4 vector (Thermo Fisher Scientific) with the native signal peptide sequence, followed by an HRV 3C cleavage site (LEVLFQGP) and the mouse IgG2b Fc region. The RAP chaperone protein (residues 1–357; GenBank: NM_002337) was cloned into the pCDNA3.4 vector. Expi293 cells (50 ml) were seeded at $1.5 × 10^6$ cells per ml, then transfected with 50 μg of LDLRAD3 and 10 μg of RAP in diluted Opti-MEM with complexed with ExpiFectamine 293 transfection reagent (Thermo Fisher Scientific). Cells were supplemented with ExpiFectamine 293 transfection enhancers 1 and 2 to boost transfection levels 1 d later. The supernatant was collected 4 d after transfection. Protein was purified using protein A Sepharose 4B (Thermo Fisher Scientific) and then dialysed into 1× HBS with 1 mM CaCl$_2$ and EDTA-free protease inhibitors (Roche). The cleaved monomeric LDLRAD3 ectodomain was obtained after incubation with HRV 3C protease (Thermo Fisher Scientific) at a 1:10 ratio overnight at 4 °C and then purified by sequential protein A Sepharose 4B and Superdex 75 size exclusion (GE Healthcare) chromatography in 20 mM HEPES pH 7.4, 150 mM NaCl and 0.01% NaN$_3$.

## Cryo-EM sample preparation, data collection and single-particle reconstruction

VEEV VLPs[33] (gift from K. Carlton and J. Mascola, Vaccine Research Center of the National Institutes of Allergy and Infectious Diseases) with and without cleaved LDLRAD3(D1) or LDLRAD3(D1+D2) in molar excess were flash-cooled on lacey carbon grids in liquid ethane using an FEI Vitrobot (Thermo Fisher Scientific). Videos of the VEEV VLPs alone and with LDLRAD3(D1) samples were recorded using the EPU software (Thermo Fisher Scientific) using a K2 Summit electron detector (Gatan) mounted onto a Bioquantum 968 GIF Energy Filter (Gatan) attached to a Titan Krios microscope operating at 300 keV with an electron dose of 35 e$^-$ Å$^{-2}$ and a magnification of ×105,000. Videos of VEEV VLPs with cleaved LDLRAD3(D1+D2) were recorded using a Falcon 4 Direct Electron Detector (Thermo Fisher Scientific) with a magnification of ×59,000. Videos from all of the samples were corrected for beam-induced motion using MotionCor2 (ref. [34]). Contrast transfer function parameters of the electron micrographs were estimated using Gctf[35], and particles were auto-picked using crYOLO[36]. Single-particle analysis, specifically reference-free 2D classification, 3D refinement, video refinement, Bayesian polishing, post-processing and local resolution estimation were performed using RELION-3.1 (ref. [37]). Post-processing of maps for model building and figure presentation was performed using DeepEMhancer[18]. Further information for all of the samples is provided in Supplementary Table 1. Structural visualization of the electron maps was performed using ChimeraX[38].

## Model building and refinement

The initial models of the VEEV structural proteins (E1, E2, transmembrane regions and capsid) with or without LDLRAD3 were constructed by docking the coordinates of the previously built model of VEEV strain TC-83 (PDB: 3J0C; ref. [21]) and the model of LDLRAD3(D1) predicted by SWISS-MODEL server[20] into the electron density of the asymmetric units of the cryo-EM maps using the fitmap command in ChimeraX. N-linked glycans and coordinated calcium ions were built manually using COOT[22]. The model underwent real-space refinement in PHENIX[23] using the default parameters plus Morphing and secondary-structure, rotamer and torsion restraints with the initial model as the reference. Bond and angle restraints were also applied for the modelled N-linked glycans and calcium ions. After optimization, coordinates of the asymmetric units were checked using MolProbity. Contact residues were identified, and buried surface areas were calculated using PDBePISA (www.ebi.ac.uk/pdbe/pisa/).

## Surface plasmon resonance

The binding kinetics and affinity of cleaved LDLRAD3(D1) or LDLRAD3(D1+D2) to VEEV VLPs were measured using the Biacore T200 system (GE Healthcare). Experiments were performed at 30 μl min$^{-1}$ and 25 °C using HBS-P (0.01 M HEPES pH 7.4, 0.15 M NaCl, 3 mM EDTA, 0.005% (v/v) Surfactant P20) plus 1 mM CaCl$_2$ as running buffer. VEEV-57 monoclonal antibody (anti-VEEV E2, N. Kafai and M. Diamond, unpublished results) was immobilized onto a CM5 sensor chip (GE Healthcare) using standard amine coupling chemistry, and VEEV VLPs were captured. LDLRAD3 proteins were injected over a range of concentrations (1 μM to 16 nM) for 300 s, followed by a 600-s dissociation period. The sensor chip was regenerated after each analyte concentration with 60 s of 10 mM glycine, pH 1.7. Before the next analyte concentrated was tested, VEEV VLPs were recaptured; the response units of captured VLPs were consistent for each cycle. All sensorgrams were double-reference-subtracted using the reference flow cell (immobilized VEEV-57 monoclonal antibody, no captured VLP) and the running-buffer blank sample. The kinetic profiles and steady-state equilibrium concentration curves were fitted using a global 1:1 binding algorithm with a drifting baseline using BIAevaluation v.3.1 (GE Healthcare).

## Infection assay

A comprehensive mutation library was generated using gene synthesis by mutating a single amino acid in D1 of the LDLRAD3 protein. The amino acids that are essential for maintaining the structural integrity of LDLRAD3 (the cysteines forming disulfide bonds, the amino acids coordinating the calcium and those forming the hydrophobic core) were kept intact[39]. The substitutions were determined using the BLOSUM scoring matrix[40] and a list of these is provided in Supplementary Table 4. The mutants were cloned into lentivirus vector pLV-EF1a-IRES-Hygro (Addgene, 85134) between the BamHI and MluI restriction enzyme sites (Genscript). An N-terminal Flag tag was added to each LDLRAD3 mutant to monitor protein expression. Δ*B4galt7*Δ*Ldlrad3* Neuro2a cells were transduced with each LDLRAD3 mutant and, 7 d later, were inoculated with SINV–VEEV (TrD)–GFP[1] (gift of W. Klimstra, University of Pittsburgh) infection at a multiplicity of infection of 20 for 7.5 h. Cells were stained with anti-Flag antibodies (1:2,000 dilution, Cell Signaling Technology, D6W5B) to measure the surface expression levels of the WT and mutant forms of LDLRAD3. Inoculated and stained cells were analysed using the MACSQuant Analyzer 10 (Miltenyi Biotec), and all flow cytometry data were processed using FlowJo (FlowJo).

## Competition binding ELISA

Nunc MaxiSorp plates (Thermo Fisher Scientific) were coated with 2 μg ml$^{-1}$ of capture monoclonal antibody (mouse anti-VEEV-1A4A)[41] in 100 μl of sodium bicarbonate coating buffer (0.1 M Na$_2$CO$_3$, pH 9.3) and incubated overnight at 4 °C. Plates were washed four times with PBS and incubated with 150 μl of blocking buffer (PBS, 4% BSA) for 1 h at room temperature. VEEV VLPs were diluted to 1 μg ml$^{-1}$ in PBS containing 2% BSA and added (100 μl per well) to plates for 1 h at room temperature. After four additional PBS washes, 50 μl of mouse anti-VEEV monoclonal antibody (3B4C-4 or TRD-14) at 20 μg ml$^{-1}$ in PBS with 2% BSA was added to plates for 30 min at room temperature to allow for binding to VEEV VLPs. Then, 50 μl of human LDLRAD3(D1)–Fc at 20 μg ml$^{-1}$ was added directly, with no additional washes. One hour later, the plates were washed four times with PBS and incubated with 100 μl per well 1:5,000 horseradish-peroxidase-conjugated goat anti-human IgG (H+L; Jackson ImmunoResearch) diluted in PBS with 2% BSA for 1 h at room temperature for detection of LDLRAD3(D1)–Fc binding. The plates were washed four times with PBS and then incubated with 100 μl of 3,3′,5,5′-tetramethylbenzidine substrate (Thermo Fisher Scientific) for 3 min at room temperature before quenching by addition of 50 μl of 2 N H$_2$SO$_4$. Absorbance was read at an optical density of 450 nm using the TriStar Microplate Reader (Berthold Technologies).

# Article

## Statistical analysis

Statistical significance was assigned when $P < 0.05$ using Prism (v.8, GraphPad) and is indicated in each of the figure legends. Cell culture or ELISA experiments were analysed using one-way ANOVA.

## Reporting summary

Further information on research design is available in the Nature Research Reporting Summary linked to this paper.

## Data availability

All data supporting the findings of this study are available within the paper and its Supplementary Information. All structures have been deposited in the PDB and Electron Microscopy Data Bank databases (PDB: 7N1I, 7N1H; EMDB: 24117, 24116, 24394). Source data are provided with this paper.

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

**Acknowledgements** We thank J. Fitzpatrick and M. Rau of the Washington University Center for Cellular Imaging and J. Errico for assistance with cryo-EM data collection and processing. We acknowledge K. Carlton and J. Mascola from the Vaccine Research Center of the National Institutes of Allergy and Infectious Diseases (NIH) for a gift of the VEEV VLPs. This study was supported by NIH grants R01AI164653 and T32AI007172 and contract HHSN272201700060C (CSGID).

**Author contributions** K.B. performed the cryo-EM reconstructions and asymmetric unit atomic modelling with support from C.A.N. and D.H.F. A.S.K. engineered and purified the LDLRAD3 proteins. H.M. performed the SINV–VEEV infection experiments. N.M.K. and S.M. designed and performed antibody inhibition studies. K.B. performed data analysis. M.S.D. and D.H.F. obtained funding for the studies. K.B., M.S.D. and D.H.F. wrote the initial manuscript draft, and the other authors provided editorial comments.

**Competing interests** M.S.D. is a consultant for Inbios, Vir Biotechnology, Fortress Biotech and Carnival Corporation and on the Scientific Advisory Board of Moderna and Immunome. The Diamond laboratory has received unrelated funding support in sponsored research agreements from Moderna, Vir Biotechnology and Emergent BioSolutions. D.H.F. is a founder of Courier Therapeutics and has received unrelated funding support from Emergent BioSolutions and Mallinckrodt Pharmaceuticals.

### Additional information

**Correspondence and requests for materials** should be addressed to Michael S. Diamond or Daved H. Fremont.

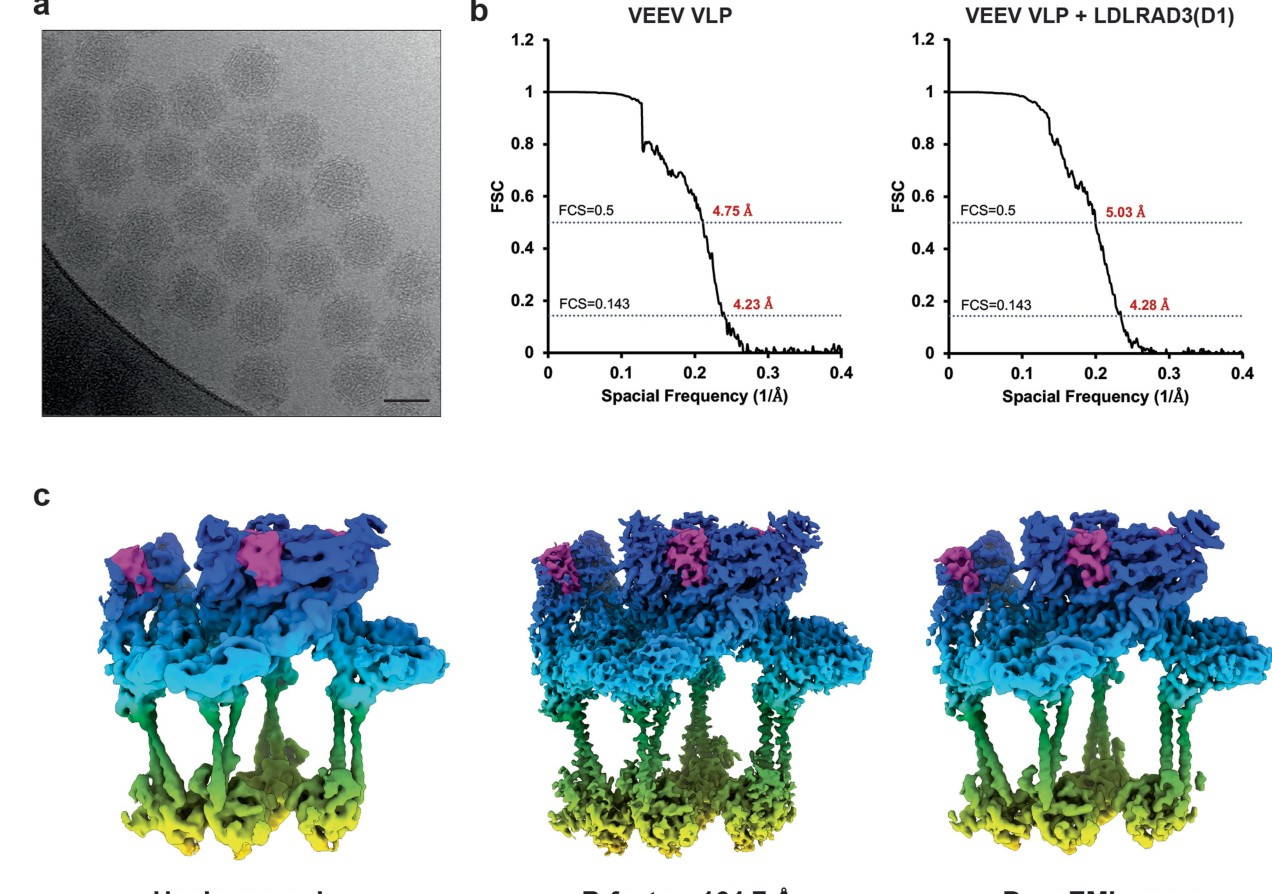

**Unsharpened**  **B-factor -164.7 Å**  **DeepEMhancer**

**Extended Data Fig. 1 | Quality assessment of cryo-EM maps.**
**a**, Representative electron micrograph (micrograph number, 1453) of VEEV VLPs. Scale bar, 500 Å. **b**, Fourier shell correlation (FSC) plots for VEEV VLPs alone (left) and with LDLRAD3(D1) (right). **c**, Side views of the unsharpened (left), globally sharpened by RELION postprocessing (middle) or modified by DeepEMhancer (right) electron densities of one asymmetric unit of VEEV–LDLRAD3 complex. The maps are coloured by radial distance from the VLP center, with LDLRAD3(D1) shown in purple, analogous to in Fig. 1b.

# Article

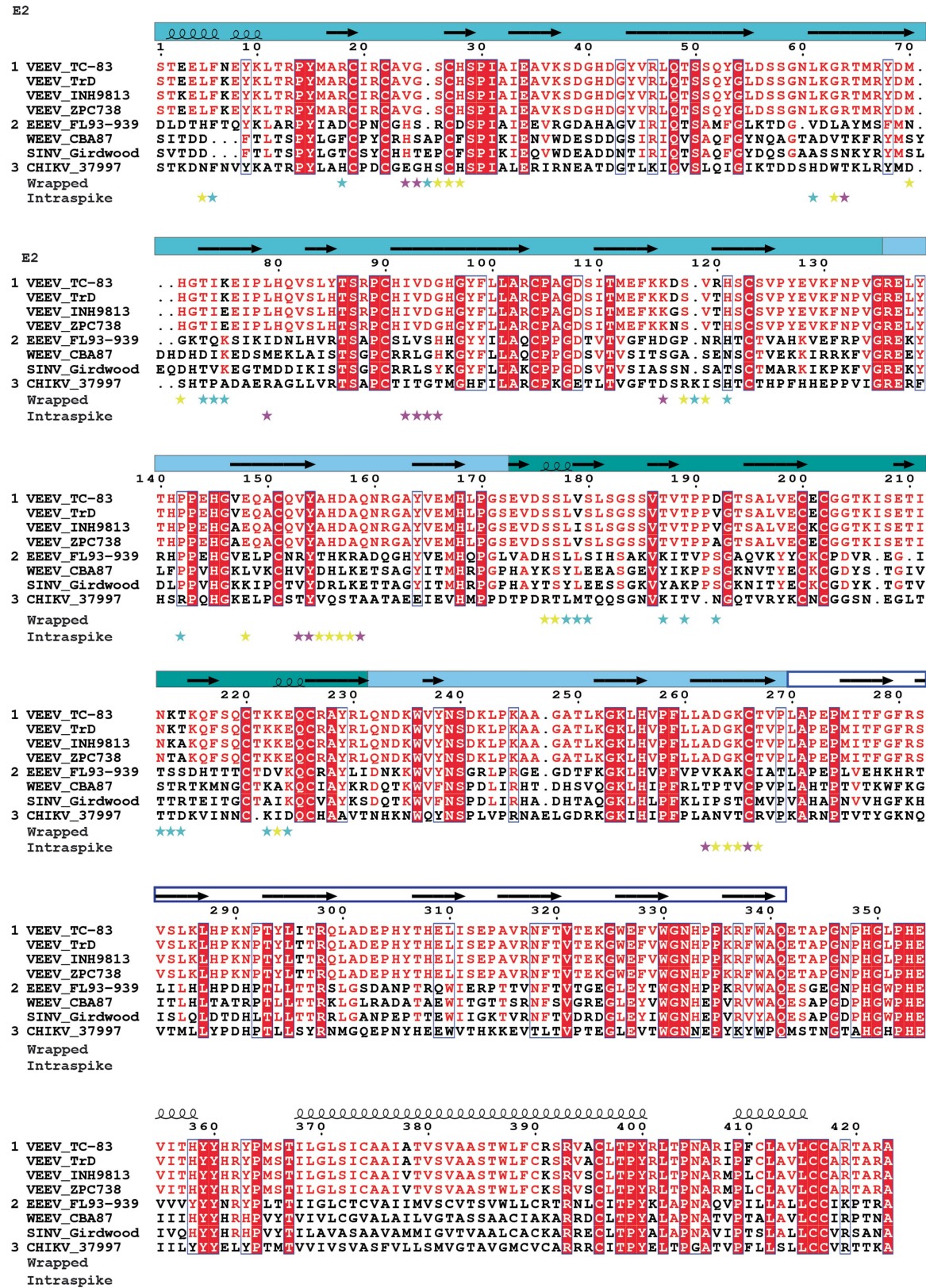

**Extended Data Fig. 2** | See next page for caption.

**Extended Data Fig. 2 | Sequence alignment of E2 proteins of the VEEV complex and other alphaviruses.** Amino acid sequence alignment of E2 proteins of various VEEV strains (IAB strain TC-83, AAB02517; IAB strain TrD, AAC19322; IC strain INH9813, AJP13627; ID strain ZPC738, AUV65225) and other alphaviruses (EEEV strain FL93-939, ABL84687; WEEV strain CBA87, ABD98014; SINV strain Girdwood, AUV65223; CHIKV strain 37997, ABX40011). Structure-based sequence alignments were performed between alphaviruses that do (group 1, left margin) or do not (groups 2 and 3, left margin) use LDLRAD3 as a receptor for infection using PROMALS3D with VEEV numbering. The figure was prepared using ESPript 3.0. Domains are coloured (A (light cyan), B (medium cyan), C (blue) and β linker (medium cyan)) and indicated above the sequence, along with the secondary structure features and nomenclature (PDB: 3J0C; ref. [21]). Red boxes indicate residues that are 100% conserved; white boxes and red letters indicate homologous residues within the specific group; white boxes and black letters indicate non-conserved residues. Determinants of receptor binding to the individual E2–E1 heterodimers are indicated by stars below the alignment and are coloured magenta if specific to LDLRAD3, cyan if specific to MXRA8, or yellow if shared between the two receptors. Wrapped denotes contacts to the wrapped E2–E1 heterodimer, the fusion loop of which is covered by LDLRAD3(D1) or MXRA8. Intraspike refers to the intraspike heterodimer, which is adjacent to the wrapped heterodimer but within the same trimeric spike. Contact residues were determined using PDBePISA.

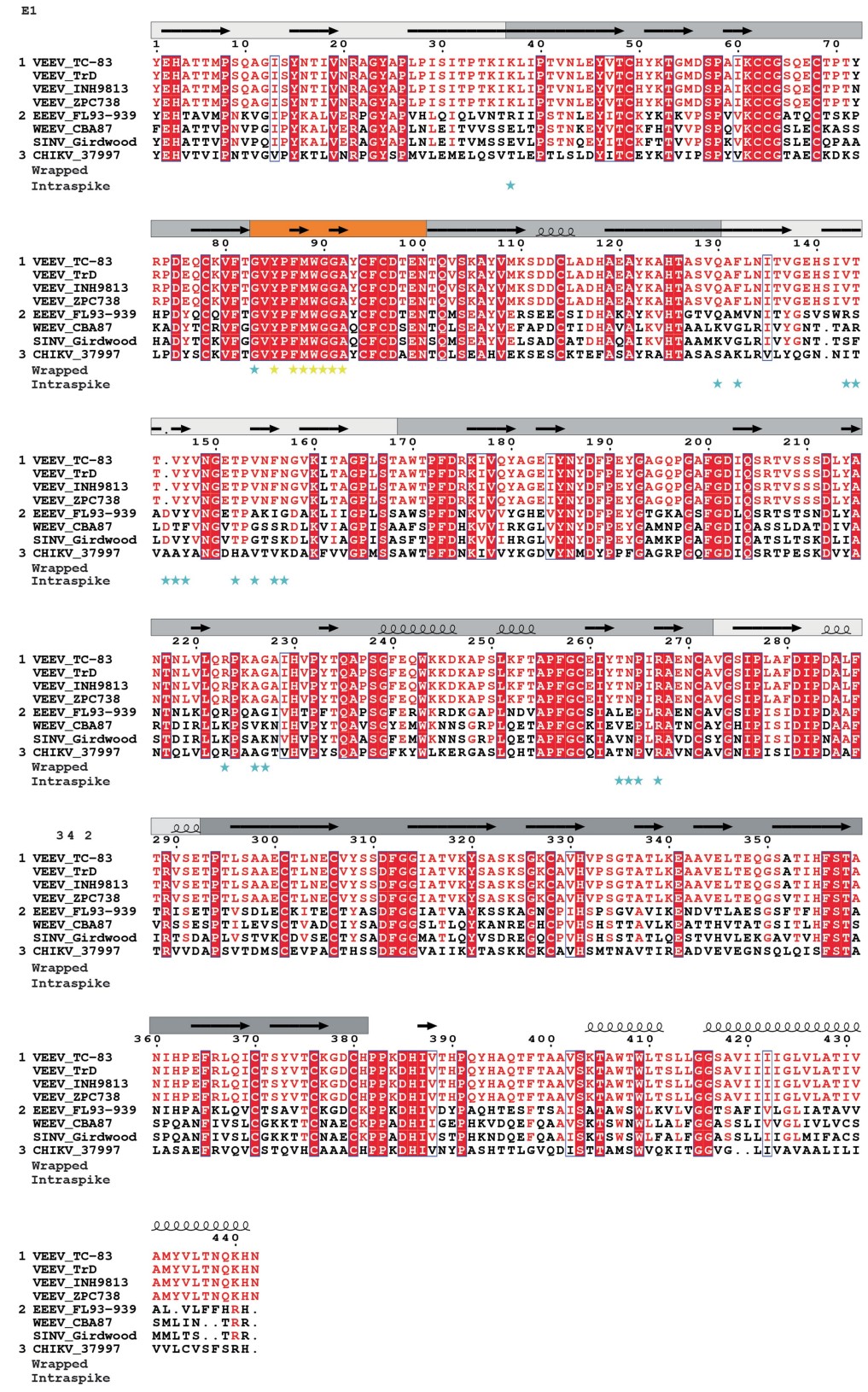

**Extended Data Fig. 3** | See next page for caption.

**Extended Data Fig. 3 | Sequence alignment of E1 proteins of the VEEV complex and other alphaviruses.** Amino acid sequence alignment of E1 proteins of various VEEV strains (IAB strain TC-83, AAB02517; IAB strain TrD, AAC19322; IC strain INH9813, AJP13627; ID strain ZPC738, AUV65225) and other alphaviruses (EEEV strain FL93-939, ABL84687; WEEV strain CBA87, ABD98014; SINV strain Girdwood, AUV65223; CHIKV strain 37997, ABX40011). Structure-based sequence alignments were performed between alphaviruses that do (group 1, left margin) or do not (groups 2 and 3, left margin) use LDLRAD3 as a receptor using PROMALS3D with VEEV numbering. The figure was prepared using ESPript 3.0. Domains are coloured (I (light grey), II (medium grey), III (dark grey) and fusion loop (orange)) and indicated above the sequence, along with the secondary structure features and nomenclature (PDB: 3J0C; ref. [21]). Red boxes indicate residues that are 100% conserved; white boxes and red letters indicate homologous residues within the specific group; white boxes and black letters indicate non-conserved residues. Determinants of receptor binding to the individual E2–E1 heterodimers are indicated by stars below the alignment and are coloured magenta if specific to LDLRAD3, cyan if specific to MXRA8 or yellow if shared between the two receptors. Wrapped denotes contacts to the wrapped E2–E1 heterodimer, the fusion loop of which is covered by LDLRAD3(D1) or MXRA8. Intraspike refers to the intraspike heterodimer, which is adjacent to the wrapped heterodimer but within the same trimeric spike. Contact residues were determined using PDBePISA.

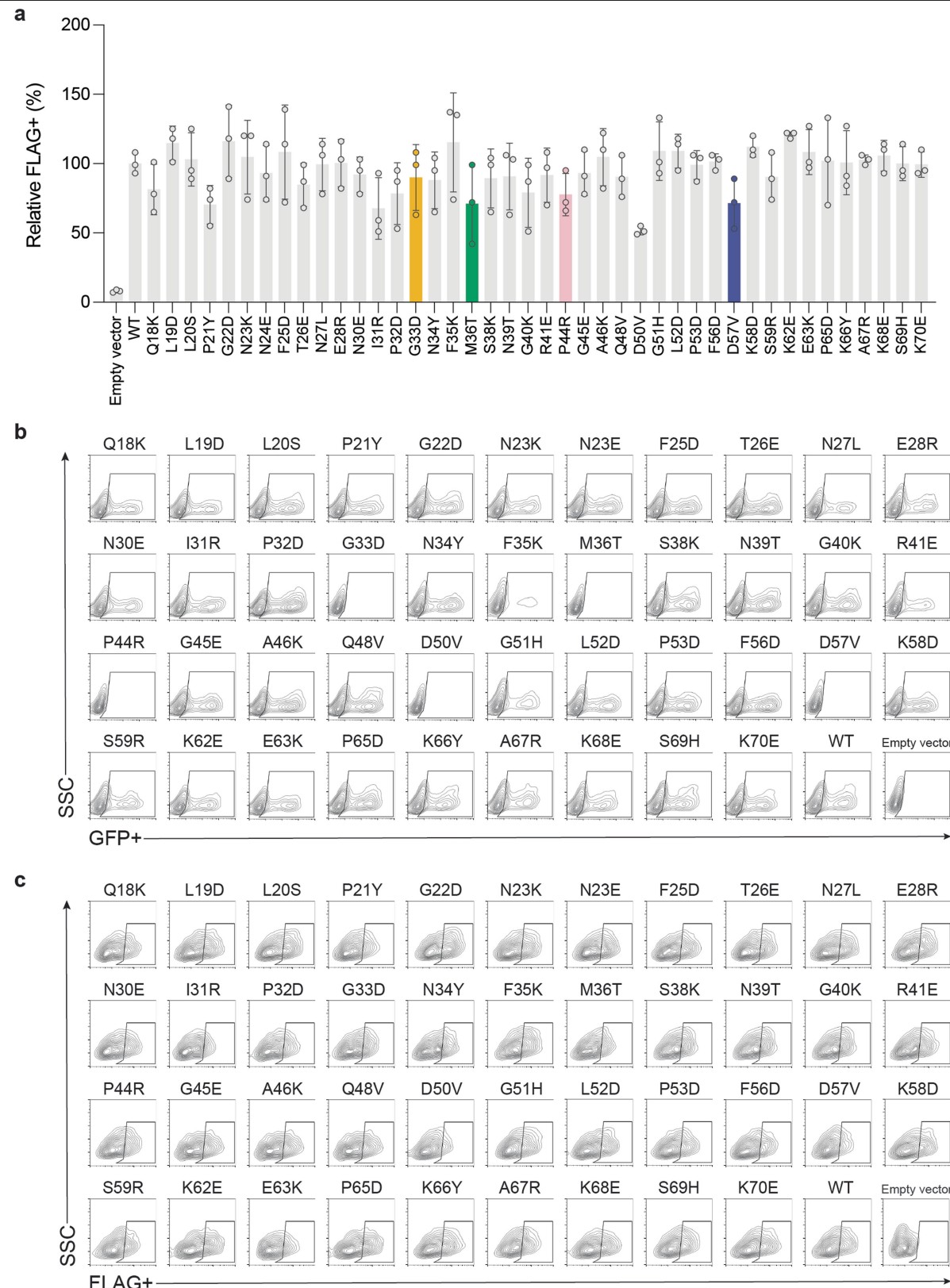

**Extended Data Fig. 4 | Expression of LDLRAD3(D1) mutants on the cell surface and the effect on SINV–VEEV infection.** Δ*B4galt7*Δ*Ldlrad3* Neuro2a cells complemented with indicated non-conservative point mutations of *Ldlrad3* in D1 (encoding an N-terminal Flag tag) were inoculated with chimeric SINV–VEEV–GFP viruses (IAB strain TrD). Then, 7.5 h later, the levels of cell surface expression of LDLRAD3 (via anti-Flag; **a**, **c**) or SINV–VEEV–GFP infection (via GFP; **b**) were assessed by flow cytometry. **a**. Data are mean ± s.d. of three experiments performed in technical duplicate. Each data symbol is the average of a technical duplicate from one experiment. **b**, **c**, Representative flow cytometry contour plots for each indicated LDLRAD3 mutant.

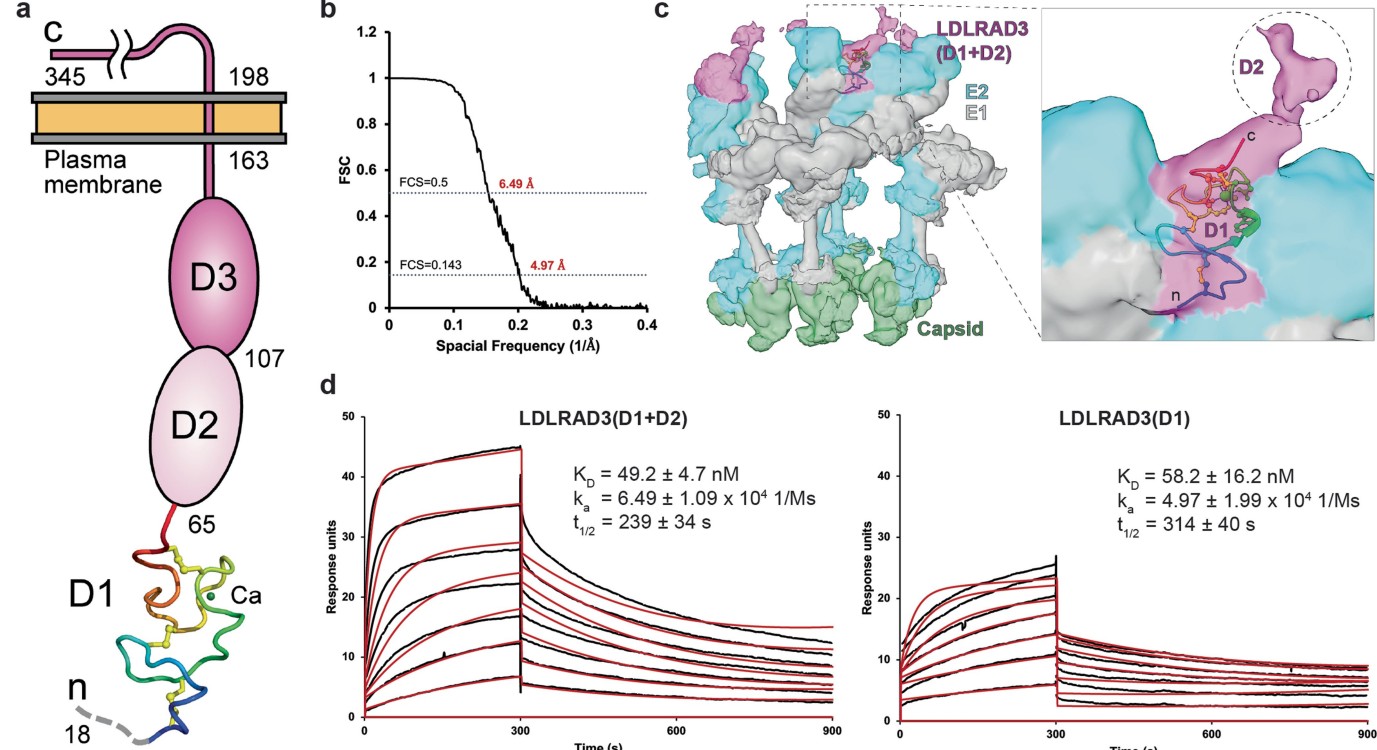

**Extended Data Fig. 5 | D2 of LDLRAD3 does not contribute to VEEV binding.**
**a**, Cartoon schematic of LDLRAD3 with labelled ectodomains and amino (n) and carboxy (c) termini. D1 is coloured in a rainbow spectrum of blue to red.
**b**, Fourier shell correlation plots for VEEV VLP with LDLRAD3(D1+D2).
**c**, Electron density of one asymmetric unit of VEEV–LDLRAD3(D1+D2) complex coloured by protein: E1 (grey), E2 (cyan), capsid (forest green) and LDLRAD3(D1+D2) (purple). Density map viewed at a low contour level to show weak density for D2 of LDLRAD3. A ribbon diagram of docked LDLRAD3(D1) model is shown with the amino to carboxy termini in a rainbow spectrum of blue to red. The cysteine residues and the acidic residues responsible for calcium ion coordination are shown as balls and sticks. The disulfide bonds and calcium ion are coloured yellow and green, respectively. Inset: magnified view of LDLRAD3 with the weak density for D2 circled. **d**, Representative surface plasmon resonance sensograms with the binding parameters of LDLRAD3(D1+D2) (left) and LDLRAD3(D1) (right) to VEEV VLPs. $n = 4$ experiments. Data are mean + s.e.m. A 1:1 binding model (red traces) was used to fit the experimental curves (black traces).

# Reporting Summary

Nature Research wishes to improve the reproducibility of the work that we publish. This form provides structure for consistency and transparency in reporting. For further information on Nature Research policies, see our Editorial Policies and the Editorial Policy Checklist.

## Statistics

For all statistical analyses, confirm that the following items are present in the figure legend, table legend, main text, or Methods section.

| n/a | Confirmed | |
|---|---|---|
| ☐ | ☒ | The exact sample size ($n$) for each experimental group/condition, given as a discrete number and unit of measurement |
| ☐ | ☒ | A statement on whether measurements were taken from distinct samples or whether the same sample was measured repeatedly |
| ☐ | ☒ | The statistical test(s) used AND whether they are one- or two-sided<br>*Only common tests should be described solely by name; describe more complex techniques in the Methods section.* |
| ☒ | ☐ | A description of all covariates tested |
| ☒ | ☐ | A description of any assumptions or corrections, such as tests of normality and adjustment for multiple comparisons |
| ☐ | ☒ | A full description of the statistical parameters including central tendency (e.g. means) or other basic estimates (e.g. regression coefficient) AND variation (e.g. standard deviation) or associated estimates of uncertainty (e.g. confidence intervals) |
| ☐ | ☒ | For null hypothesis testing, the test statistic (e.g. $F$, $t$, $r$) with confidence intervals, effect sizes, degrees of freedom and $P$ value noted<br>*Give P values as exact values whenever suitable.* |
| ☒ | ☐ | For Bayesian analysis, information on the choice of priors and Markov chain Monte Carlo settings |
| ☒ | ☐ | For hierarchical and complex designs, identification of the appropriate level for tests and full reporting of outcomes |
| ☒ | ☐ | Estimates of effect sizes (e.g. Cohen's $d$, Pearson's $r$), indicating how they were calculated |

*Our web collection on statistics for biologists contains articles on many of the points above.*

## Software and code

Policy information about availability of computer code

| | |
|---|---|
| Data collection | FEI vitrobot (Thermo Fisher), Titan Krios (FEI), K2 summit electron detector (Gatan), Falcon 4 Direct Electron Detector (Thermo Fisher), EPU (Thermo FIsher), Biacore T200 system (GE Healthcare), TriStar Microplate Reader (Berthold Technologies), MACSQuant Analyzer 10 (Miltenyi Biotec). |
| Data analysis | MotionCor2 v1.4, Gctf v1.06, crYOLO v1.7.6, RELION v3.1, DeepEMhancer v20200909 , ChimeraX v1.3 , SWISS-MODEL (https://swissmodel.expasy.org), COOT v0.9.5, PHENIX v1.19.2-4158, PDBePISA (https://www.ebi.ac.uk/pdbe/pisa), FlowJo (BD, v10.7.0), BIAevaluation v3.1 (GE Healthcare). Prism (GraphPad, v8.4.3), PROMALS3D (https://prodata.swmed.edu/promals3d), ESPript v3.0. |

For manuscripts utilizing custom algorithms or software that are central to the research but not yet described in published literature, software must be made available to editors and reviewers. We strongly encourage code deposition in a community repository (e.g. GitHub). See the Nature Research guidelines for submitting code & software for further information.

## Data

Policy information about availability of data

All manuscripts must include a data availability statement. This statement should provide the following information, where applicable:

- Accession codes, unique identifiers, or web links for publicly available datasets
- A list of figures that have associated raw data
- A description of any restrictions on data availability

The authors declare that all data supporting the findings of this study are available within the paper and its Supplementary information and are available from the corresponding author upon request. All structures are deposited in the PDB and EMDB databases (PDB 7N1I, 7N1H; EMDB 24117, 24116, 24394). GenBank sequences used in the structural alignment analysis: VEEV strain TC-83, AAB02517; VEEV strain TrD, AAC19322; VEEV strain INH9813, AJP13627; VEEV strain ZPC738, AUV65225, EEEV strain FL93-939, ABL84687; WEEV strain CBA87, ABD98014; SINV strain Girdwood, AUV65223; CHIKV strain 37997, ABX40011.

# Field-specific reporting

Please select the one below that is the best fit for your research. If you are not sure, read the appropriate sections before making your selection.

☒ Life sciences  ☐ Behavioural & social sciences  ☐ Ecological, evolutionary & environmental sciences

For a reference copy of the document with all sections, see nature.com/documents/nr-reporting-summary-flat.pdf

# Life sciences study design

All studies must disclose on these points even when the disclosure is negative.

| | |
|---|---|
| Sample size | No sample sizes were chosen a priori. All experiments were repeated at least three independent times, each with multiple technical replicates. |
| Data exclusions | No data was excluded. |
| Replication | All experiments had at least 3 independent biological replicates. All replication attempts were successful. |
| Randomization | No randomization was necessary as no human or animal subjects were used in the study. |
| Blinding | No blinding was necessary as no human or animal subjects were used inn the study. |

# Reporting for specific materials, systems and methods

We require information from authors about some types of materials, experimental systems and methods used in many studies. Here, indicate whether each material, system or method listed is relevant to your study. If you are not sure if a list item applies to your research, read the appropriate section before selecting a response.

### Materials & experimental systems

| n/a | Involved in the study |
|---|---|
| ☐ | ☒ Antibodies |
| ☐ | ☒ Eukaryotic cell lines |
| ☒ | ☐ Palaeontology and archaeology |
| ☒ | ☐ Animals and other organisms |
| ☒ | ☐ Human research participants |
| ☒ | ☐ Clinical data |
| ☒ | ☐ Dual use research of concern |

### Methods

| n/a | Involved in the study |
|---|---|
| ☒ | ☐ ChIP-seq |
| ☐ | ☒ Flow cytometry |
| ☒ | ☐ MRI-based neuroimaging |

## Antibodies

| | |
|---|---|
| Antibodies used | Anti-VEEV mAbs: 1A4A1-1, 3B4C-4, TRD-14, 57;  Rabbit anti-FLAG antibody (Cell Signaling Technology, clone D6W5B, Cat #14793S); horseradish peroxidase (HRP)-conjugated goat anti-human IgG (H+L; Jackson ImmunoResearch, Cat #109-035-003). |
| Validation | Antibodies were validated by SDS-PAGE analysis and binding to viral recombinant proteins and/or infected cells. Many of these antibodies were sequence confirmed, generated in our laboratories, and previously used for similar applications (PMID: 33208938).<br>1.  1A4A-1 (Validated by SDS-PAGE analysis and binding to VEEV-infected cells and VEEV recombinant proteins); PMID: 2414905<br>2.  3B4C-4 (Validated by SDS-PAGE analysis and binding to VEEV-infected cells and VEEV recombinant proteins); PMID: 2414905<br>3.  TRD-14 (Validated by SDS-PAGE analysis and binding to VEEV-infected cells and VEEV recombinant proteins); N.M.K. and M.S.D., unpublished<br>4.  57 (Validated by SDS-PAGE analysis and binding to VEEV-infected cells and VEEV recombinant proteins); N.M.K. and M.S.D., unpublished<br>5.  Rabbit anti-FLAG (Cell Signaling Technology, clone D6W6B, Cat #14793S); Commercially validated by flow cytometry<br>6.  Horseradish peroxidase (HRP)-conjugated goat anti-human IgG (H+L; Jackson ImmunoResearch, Cat #109-035-003); Commercially validated by ELISA |

## Eukaryotic cell lines

Policy information about cell lines

| | |
|---|---|
| Cell line source(s) | Neuro-2a (Cat #CCL-131) cells were obtained from ATCC. Expi293F (Cat #A14527) cells were obtained from Thermo Fisher. |
| Authentication | These cells were obtained from ATCC or other commercial vendors and grew and performed as expected. Morphology of each cell line was assessed by microscopy. |

| Mycoplasma contamination | All cell lines are routinely tested each month and were negative for mycoplasma. |
| Commonly misidentified lines (See ICLAC register) | This study did not involve any commonly misidentified cell lines. |

# Flow Cytometry

## Plots

Confirm that:

☒ The axis labels state the marker and fluorochrome used (e.g. CD4-FITC).

☒ The axis scales are clearly visible. Include numbers along axes only for bottom left plot of group (a 'group' is an analysis of identical markers).

☒ All plots are contour plots with outliers or pseudocolor plots.

☒ A numerical value for number of cells or percentage (with statistics) is provided.

## Methodology

| Sample preparation | After infection, cells were harvested, fixed, permeabilized (or non-permeabilized in some experiments), and stained with the antiviral or anti-FLAG antibodies described above. |
| Instrument | MACSQuant Analyzer 10 |
| Software | FlowJo software (BD) |
| Cell population abundance | rans-complemented cells were analyzed for transgene expression using anti-FLAG antibody |
| Gating strategy | Gating was performed based on non-binding control antibodies and/or uninfected cells. Dead cells were excluded by scatter and size. |

☒ Tick this box to confirm that a figure exemplifying the gating strategy is provided in the Supplementary Information.

