## [Peer Review File · Nature]

Manuscript Title: Structural basis of Venezuelan equine encephalitis virus engagement of the LDLRAD3 receptor

Reviewer Comments & Author Rebuttals

Reviewer Reports on the Initial Version:

Referees' comments:

Referee #1 (Remarks to the Author):

Basore et al describe the cryo-EM structure of the VEEV particle in complex with domain 1 of its LDLRAD3 receptor. VEEV is a mosquito-borne encephalitic alphavirus and is the cause of public health concern. The authors had recently identified LDLRAD3 as a receptor for entry into its target cells. The structure presented shows that it binds in a similar way as the unrelated MXRA8 receptor of arthritogenic alphaviruses, except that a smaller area is buried in the complex. The authors show that only domain 1 (D1) is responsible for the interactions, and that the other domains do not contribute. At a resolution of around 4Å, the details of the interaction are not provided as the side chains are not clear, but the authors provide a reasonable model of the interaction and validate the residues involved by site-directed mutagenesis, and by testing the effect of mutants in entry of a chimeric VEEV virus that can be used under BSL2 containment. Furthermore, they used two murine neutralizing antibodies of known binding site on the particle, and showed that Mab 3B4C-4, which binds at an overlapping site, competes with LDLRAD3 for binding, whereas TRD-14, which binds elsewhere and neutralizes by a different mechanism, does not.

The manuscript is very well written, the structures are of reasonable quality for the resolution achieved, and the experiments confirming the functional importance of the identified residues are convincingly described. This manuscript is thus an important contribution to understanding alphavirus biology. Given global warming and the expansion of the areas of these normally "tropical" mosquito-borne viruses, these advances should help identify molecular targets to combat the disease, and in this context the results are highly significant.

Referee #2 (Remarks to the Author):

Basore et al provide a comprehensive structure-function analysis of the interaction Venezuelan equine encephalitis virus (VEEV) with its cellular receptor, low-density lipoprotein receptor type-A (LDLRAD3) using cryoEM. They determined the structure of VEEV with and without the D1 (N-terminal ecto-domain) of LDLRAD3. The structure and interactions between D1 and the E1/E2 glycoprotein complex on VEEV are described in great detail with interacting residues well mapped on both the viral proteins and the receptor. The functional aspects of the structure are then tested with numerous mutations to the receptor in regions of virus interaction followed by binding and infectivity assays with the modified receptor in cells transfected with LDLRAD3 containing the mutated D1 domain. Most of the modified residues lead to reduced binding, although there is a possibility that some lead to enhanced binding and infectivity. An additional feature of the study is the effect of previously studied Fab fragments on the binding of VEEV to LDLRAD3 D1. The monoclonal Fab shown previously to block infection by VEEV also blocks LDLRAD3 D1 binding while another Fab that does not block infection does not inhibit binding of D1. Finally, they follow up by studying the structure and function of D1-D2 binding and show convincingly that domain 2 plays little or no role in virus binding. In addition to the virus study with VEEV, there are interesting

comparisons with previously published work on the alpha Chikungunya virus and its receptor binding with MXRA8. In fact, the two receptors bind in a similar manner to the tertiary structures of the glycoproteins from two different trimers of the two different viruses even though the receptors are completely different molecules and there are widely differing sequences in the glycoproteins in the two different viruses.

The paper makes an important contribution to the virus-receptor interaction literature with the added feature, as suggested by the authors, that there may be small molecules that can have a broader than expected effect on virus attachment due to the similar binding features of different receptors to the different alphavirus glycoproteins. The structural work appears to be of the highest quality and, to the extent of my understanding, so are the functional studies. In my opinion the discussion and experiments with the Fab fragments may actually dilute the impact of the other structural results since there was nothing unexpected in this outcome and it made the paper drag on to some extent. The paper should be of broad interest to the virology community.

I was not able to find any reference to the natural binding partner to LDLRAD3 and of course that would be of great interest. Recently (Su C, Wu L, Chai Y, Qi J, Tan S, Gao GF, Song H, Yan J. Molecular basis of EphA2 recognition by gHgL from gammaherpesviruses. Nat Commun. 2020 Nov 24;11(1):5964. doi: 10.1038/s41467-020-19617-9. PMID: 33235207; PMCID: PMC7687889) it was shown that the EphA2 receptor for HH8 bound to the viral glycoprotein in the same manner as its natural binding partner, suggesting that signaling may also be a role in addition to attachment. A few words about such a possibility might be of interest in the discussion if there is anything known about the natural downstream signaling of LDLRAD3.

Author Rebuttals to Initial Comments:

Response to Referee Comments

Referee #1.

Basore et al describe the cryo-EM structure of the VEEV particle in complex with domain 1 of its LDLRAD3 receptor. VEEV is a mosquito-borne encephalitic alphavirus and is the cause of public health concern. The authors had recently identified LDLRAD3 as a receptor for entry into its target cells. The structure presented shows that it binds in a similar way as the unrelated MXRA8 receptor of arthritogenic alphaviruses, except that a smaller area is buried in the complex. The authors show that only domain 1 (D1) is responsible for the interactions, and that the other domains do not contribute. At a resolution of around 4Å, the details of the interaction are not provided as the side chains are not clear, but the authors provide a reasonable model of the interaction and validate the residues involved by site-directed mutagenesis, and by testing the effect of mutants in entry of a chimeric VEEV virus that can be used under BSL2 containment. Furthermore, they used two murine neutralizing antibodies of known binding site on the particle, and showed that Mab 3B4C-4, which binds at an overlapping site, competes with LDLRAD3 for binding, whereas TRD-14, which binds elsewhere and neutralizes by a different mechanism, does not.

The manuscript is very well written, the structures are of reasonable quality for the resolution achieved, and the experiments confirming the functional importance of the identified residues are convincingly described. This manuscript is thus an important contribution to understanding alphavirus biology. Given global warming and the expansion of the areas of these normally “tropical” mosquito-borne viruses, these advances should help identify molecular targets to combat the disease, and in this context the results are highly significant.

We greatly appreciated the positive summary and comments.

Referee #2.

Basore et al provide a comprehensive structure-function analysis of the interaction Venezuelan equine encephalitis virus (VEEV) with its cellular receptor, low-density lipoprotein receptor type-A (LDLRAD3) using cryoEM. They determined the structure of VEEV with and without the D1 (N-terminal ecto-domain) of LDLRAD3. The structure and interactions between D1 and the E1/E2 glycoprotein complex on VEEV are described in great detail with interacting residues well mapped on both the viral proteins and the receptor. The functional aspects of the structure are then tested with numerous mutations to the receptor in regions of virus interaction followed by binding and infectivity assays with the modified receptor in cells transfected with LDLRAD3 containing the mutated D1 domain. Most of the modified residues lead to reduced binding, although there is a possibility that some lead to enhanced binding and infectivity. An additional feature of the study is the effect of previously studied Fab fragments on the binding of VEEV to LDLRAD3 D1. The monoclonal Fab shown previously to block infection by VEEV also blocks LDLRAD3 D1 binding while another Fab that does not block infection does not inhibit binding of D1. Finally, they follow up by studying the structure and function of D1-D2 binding and show convincingly that domain 2 plays little or no role in virus binding. In addition to the virus study with VEEV, there are interesting comparisons with previously published work on the alpha Chikungunya virus and its receptor binding with MXRA8. In fact, the two receptors bind in a similar manner to the tertiary structures of the glycoproteins from two different trimers of the two different viruses even though the receptors are completely different molecules and there are widely differing sequences in the glycoproteins in the two different viruses.

The paper makes an important contribution to the virus-receptor interaction literature with the added feature, as suggested by the authors, that there may be small molecules that can have a broader than expected effect on virus attachment due to the similar binding features of different receptors to the different alphavirus glycoproteins. The structural work appears to be of the highest quality and, to the extent of my understanding, so are the functional studies. In my opinion the discussion and experiments with the Fab fragments may actually dilute the impact of the other structural results since there was nothing unexpected in this outcome and it made the paper drag on to some extent. The paper should be of broad interest to the virology community.

We also greatly appreciated these favorable comments.

I was not able to find any reference to the natural binding partner to LDLRAD3 and of course that would be of great interest. Recently (Su C, Wu L, Chai Y, Qi J, Tan S, Gao GF, Song H, Yan J. Molecular basis of EphA2 recognition by gHgL from gammaherpesviruses. Nat Commun. 2020 Nov 24;11(1):5964. doi: 10.1038/s41467-020-19617-9. PMID: 33235207; PMCID: PMC7687889) it was shown that the EphA2 receptor for HH8 bound to the viral glycoprotein in the same manner as its natural binding partner, suggesting that signaling may also be a role in addition to attachment. A few words about such a possibility might be of interest in the discussion if there is anything known about the natural downstream signaling of LDLRAD3.

We thank the referee for the suggestion. While still poorly characterized, LDLRAD3 contains two conserved polyproline motifs within its cytoplasmic domain that may interact with the

WW- domain-containing proteins often involved in cell signaling pathways. LDLRAD3 was shown to interact in this manner to the E3 ubiquitin ligase Itch, leading to its auto-ubiquitination and degradation (Noyes et al., 2016). Although the binding interaction remains undefined, LDLRAD3 also reportedly associates with amyloid precursor protein and modulates its cellular processing (Ranganathan et al., 2011). Since binding partners remain unclear for LDLRAD3, particularly to the ectodomain, we are hesitant to make speculative statements on its biological role. We have, however, added the following brief clause in the introduction:

“LDLRAD3 is a conserved yet poorly characterized cell surface protein expressed in neurons, epithelial cells, myeloid cells, and muscle tissue whose endogenous ligand(s) remain unknown”.

References Cited:

Noyes, N. C., Hampton, B., Migliorini, M. & Strickland, D. K. Regulation of itch and Nedd4 E3 ligase activity and degradation by LRAD3. *Biochemistry* **55**, 1204–1213 (2016).

Ranganathan, S. *et al.* LRAD3, a novel low-density lipoprotein receptor family member that modulates amyloid precursor protein trafficking. *J. Neurosci.* **31**, 10836–10846 (2011).